# Mass media exposure and self-efficacy in abortion decision-making among adolescent girls and young women in Ghana: Analysis of the 2017 Maternal Health Survey

Bright Opoku Ahinkorah[1], Abdul-Aziz Seidu[2,3], Georgina Yaa Mensah[4], Eugene Budu[2]*

1 Faculty of Health, The Australian Centre for Public and Population Health Research (ACPPHR), University of Technology Sydney, Ultimo, Australia, 2 Department of Population and Health, University of Cape Coast, Cape Coast, Ghana, 3 College of Public Health, Medical and Veterinary Services, James Cook University, Townsville, Australia, 4 Department of Communication Studies, University of Cape Coast, Cape Coast, Ghana

* budueugene@gmail.com

**Data Availability Statement:** The dataset is freely available to the public at https://dhsprogram.com/data/dataset/Ghana_Special_2017.cfm?flag=0.

## Abstract

### Introduction

Worldwide, 25 million unsafe abortions (45% of all abortions) occurred every year between 2010 and 2014 and 97%, occurred in low-and-middle income countries. Abortion among adolescent girls and young women (15–24 years) is a major public health issue, especially in low-and middle-income countries, including Ghana. Using data from the 2017 Maternal Health Survey, we sought to examine the association between mass media exposure and adolescent girls and young women's self-efficacy in abortion decision making.

### Materials and methods

A sample of 5,664 adolescent girls and young women in Ghana was considered in this study. Both descriptive and inferential analytical approaches were employed to analyse the data. The descriptive analytical approach involved the use of proportions to illustrate the proportion of adolescent girls and young women who had self-efficacy in abortion decision-making. Self-efficacy in abortion decision-making was derived from the question 'Could you decide on your own to get an abortion?' Respondents who answered "Yes" to this question were considered as having self-efficacy in abortion decision making. At the inferential level, a chi-square test and bivariate and multivariable logistic regression models were employed with statistical significance pegged at p-value <0.05. The results of the bivariate and multivariable logistic regression analyses were presented using crude and adjusted odds ratios respectively.

### Results

Less than a quarter of adolescent girls and young women (24%) in Ghana had self-efficacy in abortion decision-making. We further found that adolescent girls and young women who

**Funding:** The author(s) received no specific funding for this work.

**Competing interests:** The authors have declared that no competing interests exist.

were exposed to mass media had higher odds in self-efficacy in abortion decision-making compared to those who were not exposed to the mass media [AOR = 1.55, CI = 1.14–2.11]. It was also found that adolescent girls and young women aged 20–24 [AOR = 1.45, CI = 1.25–1.68], those who were cohabiting [AOR = 1.40, CI = 1.02–1.93], and those from the Ashanti region [AOR = 2.39, CI = 1.85–3.07] had higher odds on self-efficacy in abortion decision-making. On the other hand, adolescent girls and young women from the Eastern Region [AOR = 0.52, CI = 0.36–0.73] and those belonging to the Ga-Adangbe ethnic group [AOR = 0.70, CI = 0.50–0.99] had lower odds in self-efficacy in abortion decision-making.

## Conclusion

Less than a quarter of adolescent girls and young women in Ghana have self-efficacy in abortion decision-making which can affect adolescent girls and young women's future abortion seeking behaviours. Exposure to mass media was strongly associated with self-efficacy in abortion decision making. We recommend that policy makers should promote mass media campaigns scheduled on regular intervals in order to inform the target audience about safe abortions in Ghana. This could go a long way to ensure that cases of unsafe abortions are reduced to the starkest minimum.

## Introduction

Abortion among adolescent girls and young women (15–24 years) is a major public health issue, especially in low-and middle-income countries [1–3]. Worldwide, 25 million unsafe abortions (45% of all abortions) occurred every year between 2010 and 2014 and 97%, occurred in low-and-middle income countries [4]. According to the Guttmacher Institute [5], 2.5 million unsafe abortions occur each year, in sub-Saharan Africa (SSA) among adolescents aged 15–19. In Ghana, of 1880 women aged 15–49 years who induced abortion in the period 2012–2017, 64% of them had an unsafe induced abortion [6].

The high prevalence of unsafe induced abortion can be attributed to the societal perception that 'abortion is a sin' and thus, those who seek and those who provide abortion services are often stigmatised [7]. This makes it difficult for women, especially adolescent girls and young women, to openly seek for abortion services and this affects their abortion decision-making.

One of the key contexts to understand women's abortion decision making is their abortion-specific experiences which may include their access to health facilities for pregnancy testing and relevant diagnosis and their ability to disclose the information to others such as partner, family and friends with attendant implications [8]. Such experiences are often influenced by their self-efficacy [9]. Self-efficacy may be described as the belief we have in our own abilities, specifically our ability to meet the challenges ahead of us and complete a task successfully [10]. In relation to abortion decision-making, adolescent girls and young women who feel that they are able to take sole decisions on their own reproductive health may be more likely to accomplish these goals. Self-efficacy may therefore, be relevant for predicting adolescent girls and young women's future abortion-seeking behaviours [9].

Over the past few decades, media campaigns have been used as an attempt to enhance various health behaviours in mass populations [11]. Typical campaigns have placed messages in media that target audiences, most frequently through television or radio, but also outdoor media, such as billboards and posters, and print media, such as magazines and newspapers [12]. Exposure to such messages is generally passive, resulting from an incidental effect of

routine use of media. Some campaigns incorporate new technologies (eg, the internet, mobile phones and personal digital assistants) [11]. Like any other health information, information on abortion can be accessed through the media [13,14]. In Ghana, media campaigns such as "TimetoTalkGH" "Smash abortion stigma" have helped to enhance the knowledge of women on safe abortion and improved their reproductive health in general [15].

With the advent of internet, it is expected that a lot more of adolescent girls and young women in Ghana will have information on abortion laws and services in the country. Whether or not exposure to media such as newspaper, radio, television and internet improves the self-efficacy in abortion decision-making of adolescent girls and young women in Ghana or not is unknown. Using data from the 2017 Ghana Maternal Health Survey (GMHS), we examined the association between mass media exposure and decision-making regarding abortion among adolescent girls and young women in Ghana. We hypothesized that adolescent girls and young women who have been exposed to media are able to decide on their own to have an induced abortion.

## Materials and methods

### Data source

Data for the study was obtained from the 2017 GMHS. It is the second nationally representative household survey after the 2007 GMHS that collects comprehensive information on maternal health issues, maternal mortality, and specific causes of death among women in Ghana. The survey gathered information on maternal health using two phases. In the first phase, 900 enumeration areas (EAs) (466 in urban areas and 434 in rural areas) were selected. The second phase involved an interview of 6,324 households, and in these households, 25,062 eligible women, aged 15–49 were asked questions about a wide range of maternal health-related issues including pregnancies, live births, abortions and miscarriages, and utilization of health services about these events. The goals of the 2017 GMHS were to collect nationally representative data that will allow an assessment of the level of maternal mortality in Ghana for the country as a whole and the Coastal, Middle, and Northern zones and identify specific causes of maternal and non-maternal deaths. Based on these goals, the survey sought to collect data on women's perceptions of and experiences with antenatal, maternity, and emergency obstetrical care, especially with regard to care received before, during, and following the termination of a pregnancy, and to measure indicators of the utilization of maternal health services, especially post-abortion care services [16]. For this study, only adolescent girls and young women (15–24 years) were considered. The total sample of eligible adolescent girls and young women with complete information in the dataset was 5,664. Hence, the sample size used in this study was 5,664. In this study, we relied on the Strengthening the Reporting of Observational Studies in Epidemiology' (STROBE) statement in writing the manuscript. The dataset is freely available to the public at https://dhsprogram.com/data/dataset/Ghana_Special_2017.cfm?flag=0.

### Measurement of variables

**Dependent variable.** The dependent variable for the study was self-efficacy in abortion decision-making. Self-efficacy in abortion decision-making was derived from the question 'Could you decide on your own to get an abortion? [9]. Those who answered "Yes" to this question were considered as having self-efficacy in abortion decision making whiles those who responded "No" to the question were considered as not having self-efficacy in abortion decision making. These responses were coded as 1 = "Yes" and 0 = "No".

**Independent variable.**    The main independent variable for the study was mass media exposure (Radio, Newspaper, Television and Internet). In the survey, respondents were asked about their frequency of listening to radio, reading newspaper and watching television. The responses were 'almost every day, at least once a week, less than once a week or not at all'. They were also asked if they had ever used the internet. The responses to this question was 'yes and no'. For the purpose of the study and ease of analysis, the responses to the questions on frequency of listening to radio, reading newspaper and watching television were recoded as follows: almost every day, at least once a week, less than once a week = 1 and not at all = 0. The responses to the question on internet use was also coded as yes = 1 and no = 0. Hence, a dichotomous outcome of 'yes and no' was created from these four questions. Any respondent who selected at least one 'yes' for all the four was considered as exposed to mass media whiles those who selected 'no' for all the four questions were considered as not-exposed to mass media [17].

**Covariates.**    Apart from mass media exposure, eight other covariates were considered in this study. These were age, level of education, place of residence, marital status, region, religion, ethnicity, and parity. The choice of the variables was influenced by their availability in the GMHS dataset and previous studies that found these variables to be important variables influencing self-efficacy in abortion decision-making [9,18]. Three of these variables (religion, ethnicity and parity) were recoded. Religion was recoded as no religion, Christian, Muslim, and Traditionalist. Ethnicity was categorized into Akan, Ga-Dangme, Ewe, Mole-Dagbani and Others. Finally, parity was recoded into zero births, one birth, two births and three or more births.

## Statistical analyses

This study employed both descriptive and inferential analytical approaches to analyze data. The descriptive analytical approach involved the use of proportions to illustrate the proportion of adolescent girls and young women who could decide on their own to get an abortion and distribution of self-efficacy in abortion decision-making across the various socio-demographic characteristics of the respondents, including mass media exposure. With the inferential analytical approach, a chi-square test ($\chi^2$) was first used to cross-tabulate the survey data results and find the association between the independent variables and the dependent variable. Apart from religion and parity, all the independent variables showed statistically significant relationship with the outcome variable and were included in a Binary Logistic Regression model to assess their influence on self-efficacy in abortion decision-making. Two regression models were developed. Model I looked at the influence of mass media exposure on self-efficacy in abortion decision-making and Model II looked at the influence of mass media exposure and all the covariates on self-efficacy in abortion decision-making. The study employed Binary Logistic Regression because this technique allows extrapolation on a combination of continuous and categorical variables [19]. The data was weighted with the available sample weight factor within the GMHS dataset to subside the effect of sampling bias and ensure generalization [6,20]. STATA version 14.0 was used to do the analyses. The coefficients of the models were exponentiated to derive Crude Odds Ratios (CORs) and Adjusted Odds Ratios (AORs). Statistically significant results were assessed at 95% confidence level.

## Ethical statement

Since the study involved the participation of human subjects, the ICF Institutional Review Board (IRB) approved the protocol for the 2017 GMHS. Nonetheless, since the researchers were not directly involved in the data collection, no ethical clearance was sought for this

particular study. However, permission was obtained from ICF Macro for the use of the dataset in this study and the terms of use have been strictly observed.

## Results

### Self-efficacy in abortion decision-making

Less than a quarter of adolescent girls and young women (24%) in Ghana could decide on their own to get an abortion (Fig 1).

### Self-efficacy in abortion decision-making across mass media exposure and socio-demographic characteristics

Adolescent girls and young women in Ghana who are able to take decisions on their own to get abortion are mostly those who have media exposure (24.3%), those with tertiary level of education (35.3%), those aged 20–24 (28.6%), those cohabiting (30.6%%), Christians (24.3%), Akans (29.2%), those from the Ashanti Region (39.1%), urban dwellers (25.7%) and those with one birth (27.0%). The results further show that media exposure ($\chi^2$ = 19.7, p< 0.001), educational level ($\chi^2$ = 49.8, p< 0.001), age ($\chi^2$ = 53.2, p< 0.001), marital status ($\chi^2$ = 14.7, p< 0.01), ethnicity ($\chi^2$ = 70.4, p< 0.001), region ($\chi^2$ = 280.4, p< 0.001) and place of residence ($\chi^2$ = 18.2, p< 0.001) had statistically significant associations with self-efficacy in abortion decision-making among adolescent girls and young women in Ghana (Table 1).

### Influence of mass media exposure and socio-demographic characteristics on self-efficacy in abortion decision-making among adolescent girls and young women in Ghana

Mass media exposure alone had a statistically significant influence on self-efficacy in abortion decision-making among adolescent girls and young women in Ghana, with adolescent girls

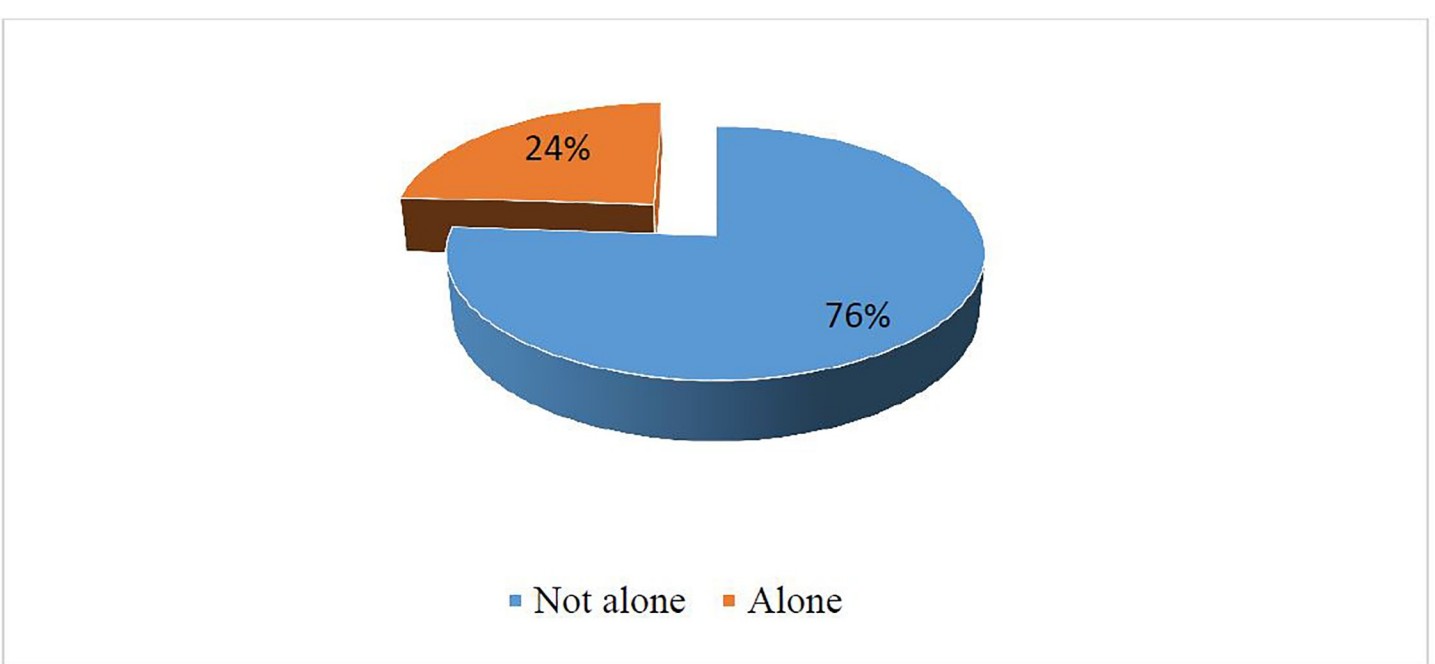

**Fig 1. Self-efficacy in abortion decision-making.** Source: Computed from 2017 GMHS.

**Table 1. Self-efficacy in abortion decision-making across mass media exposure and socio-demographic characteristics (N = 5,664).**

| Variable | Weighted N | Percentage | Self-efficacy in abortion decision-making | $X^2$ (p-value) |
|---|---|---|---|---|
| **Mass media Exposure** | | | | 19.7 (p< 0.001) |
| Not exposed | 246 | 4.0 | 14.5 | |
| Exposed | 5,417 | 96.0 | 24.3 | |
| **Educational level** | | | | 49.8 (p< 0.001) |
| No education | 10 | 0.2 | 22.2 | |
| Primary | 346 | 6.1 | 13.8 | |
| Secondary | 4,896 | 86.5 | 23.4 | |
| Tertiary | 412 | 7.3 | 35.3 | |
| **Age** | | | | 53.2 (p< 0.001) |
| 15–19 | 3,388 | 59.8 | 20.2 | |
| 20–24 | 3,376 | 40.2 | 28.6 | |
| **Marital Status** | | | | 14.7 (p< 0.01) |
| Married | 275 | 4.9 | 21.7 | |
| Cohabiting | 564 | 10.0 | 30.6 | |
| Single | 4,825 | 85.2 | 23.1 | |
| **Religion** | | | | 5.7(p = 0.126) |
| No religion | 52 | 0.9 | 20.9 | |
| Christian | 4,768 | 84.2 | 24.3 | |
| Muslim | 805 | 14.2 | 21.6 | |
| Traditionalist | 39 | 0.7 | 14.7 | |
| **Ethnicity** | | | | 70.4 (p< 0.001) |
| Akan | 2,780 | 49.4 | 29.2 | |
| Ga-Adangbe | 536 | 9.5 | 17.6 | |
| Ewe | 823 | 14.5 | 15.6 | |
| Mole-Dagbani | 766 | 13.5 | 22.2 | |
| Others | 738 | 13.0 | 21.1 | |
| **Region** | | | | 280.4 (p< 0.001) |
| Western | 733 | 13.0 | 20.5 | |
| Central | 495 | 8.8 | 37.0 | |
| Greater Accra | 1,132 | 20.0 | 23.7 | |
| Volta | 470 | 8.3 | 11.1 | |
| Eastern | 610 | 10.8 | 11.0 | |
| Ashanti | 1,144 | 20.2 | 39.1 | |
| Brong Ahafo | 486 | 8.6 | 29.2 | |
| Northern | 265 | 4.7 | 18.2 | |
| Upper East | 197 | 3.5 | 14.6 | |
| Upper West | 132 | 2.3 | 28.7 | |
| **Place of Residence** | | | | 18.2 (p< 0.001) |
| Urban | 3,376 | 59.6 | 25.7 | |
| Rural | 2,288 | 40.4 | 20.9 | |
| **Parity** | | | | 5.9 (p = 0.118) |
| No births | 4,651 | 82.1 | 23.1 | |
| One birth | 776 | 13.7 | 27.0 | |
| Two births | 227 | 4.0 | 23.8 | |
| Three or more births | 10 | 0.2 | 14.3 | |

Source: Computed from 2017 GMHS.

and young women who were exposed to mass media more likely to take decisions on their own to get an abortion compared to those who were not exposed to media as shown in Model I [COR = 1.90, CI = 1.43–2.53]. By interacting with the socio-demographic characteristics of adolescent girls and young women, mass media exposure, together with age, marital status, ethnicity and region showed statistically significant influence on self-efficacy in abortion decision-making among adolescent girls and young women in Ghana (see Model II). With media exposure, adolescent girls and young women who were exposed to media were more likely to take decisions on their own to get an abortion compared to those who were not exposed to media [AOR = 1.55, CI = 1.14–2.11]. With age, adolescent girls and young women aged 20–24 [AOR = 1.45, CI = 1.25–1.68] were more likely to take decisions on their own to get an abortion compared to those aged 15–19. The likelihood of taking decisions on their own to get an abortion was high among adolescent girls and young women who were cohabiting [AOR = 1.40, CI = 1.02–1.93] in terms of marital status. With region, the highest likelihood of self-efficacy in abortion decision making was found among adolescent girls and young women from the Ashanti Region [AOR = 2.39, CI = 1.85–3.07] whiles those from the Eastern Region were less likely to take their own decisions to get an abortion [AOR = 0.52, CI = 0.36–0.73]. Finally, the likelihood of self-efficacy in abortion decision making among adolescent girls and young women in Ghana was low among those belonging to the Ga-Adangbe ethnic group [AOR = 0.70, CI = 0.50–0.99] (Table 2).

## Discussion

In this study, we used data from the 2017 GMHS to examine the association between mass media exposure and self-efficacy in abortion decision-making among adolescent girls and young women in Ghana. We hypothesized that adolescent girls and young women who have been exposed to mass media would have high self-efficacy in abortion decision-making compared to those who are not exposed to mass media. The study revealed that only 24% of adolescent girls and young women in Ghana have self-efficacy in abortion decision-making. This is consistent with a previous study that used the same dataset to examine the abortion experiences among women in Ghana [9].

Arguably, mass media has permeated every sphere of life among adolescent girls and young people in this 21st century [21]. This study also showed that adolescent girls and young women who are exposed to mass media are more likely to have self-efficacy to make abortion decisions. This corroborates other studies on the association between mass media exposure and reproductive health decision-making capacities and behaviour in Ghana [9] and other parts of SSA [22]. Specifically, Owoo, Lambon-Quayefio and Onuoha, [9] found that mass media exposure has strong association with abortion self-efficacy among women in their reproductive age. The probable explanation the authors gave to this association was that women who are exposed to mass media are more likely to be exposed to *"western ideas and messages that encourage independence and greater autonomy"*. In addition to this, the various channels such as the internet that have permeated the life of most young people could be used to find information on the various locations and means by which young people could employ to terminate a pregnancy.

Aside the principal independent variable–mass media exposure, the association of the covariates with the outcome variable are worth discussing. The likelihood of self-efficacy in abortion decision-making was high among adolescent girls and young women who were cohabiting and those aged 20–24. This is consistent with previous studies on the association between age and decision-making on sexual and reproductive health services. For instance, the study by Seidu et al. [23] found that women in advanced ages compared to adolescents were

**Table 2. Influence of mass media exposure and socio-demographic characteristics on self-efficacy in abortion decision-making among adolescent girls and young women in Ghana.**

| Variable | Model I COR (95% CI) | Model II AOR (95% CI) |
|---|---|---|
| **Mass Media Exposure** | | |
| Not exposed | Ref | Ref |
| Exposed | 1.90***[1.43–2.53] | 1.55**[1.14–2.11] |
| **Age** | | |
| 15–19 | | Ref |
| 20–24 | | 1.45***[1.25–1.68] |
| **Educational level** | | |
| No education | | Ref |
| Primary | | 0.86[0.16–4.64] |
| Secondary | | 1.32[0.25–7.0] |
| Tertiary | | 1.75[0.33–9.44] |
| **Marital Status** | | |
| Married | | Ref |
| Cohabiting | | 1.40*[1.02–1.93] |
| Single | | 1.14[0.87–1.49] |
| **Ethnicity** | | |
| Akan | | Ref |
| Ga-Adangbe | | 0.70*[0.50–0.99] |
| Ewe | | 0.83[0.62–1.13] |
| Mole-Dagbani | | 0.96[0.77–1.20] |
| Others | | 0.93[0.75–1.16] |
| **Region** | | |
| Western | | Ref |
| Central | | 2.25***[1.67–3.02] |
| Greater Accra | | 1.28[0.94–1.73] |
| Volta | | 0.56*[0.35–0.89] |
| Eastern | | 0.52***[0.36–0.73] |
| Ashanti | | 2.39***[1.85–3.07] |
| Brong Ahafo | | 1.64***[1.22–2.19] |
| Northern | | 0.94[0.67–1.30] |
| Upper East | | 0.72[0.51–1.01] |
| Upper West | | 1.74**[1.26–2.42] |
| **Place of residence** | | |
| Urban | | Ref |
| Rural | | 0.95[0.82–1.09] |

Exponentiated coefficients; 95% confidence intervals in brackets, COR = Crude Odds Ratio, AOR = Adjusted Odds Ratio, Ref = Reference

* $p < 0.05$

** $p < 0.01$

*** $p < 0.001$.

Source: Computed from 2017 GMHS.

more likely to make decisions in relations to their reproductive health including abortions. There are two principal reasons that may explain this association. First, there are cultural connotations that expect younger people to be obedient and seek advice from the elderly in all situations. In Africa and Ghana in particular, age is very important in decision-making processes

in that younger persons are expected to be submissive when it comes to decision making with the older ones [24]. It has also been indicated that being an adolescent woman in sexual unions may result in powerlessness and difficulty in reproductive health decision-making [24,25]. The second pathway may be explained by the legal age that women in Ghana can seek abortion depending on the reasons stipulated in the abortion law of the country. Young people who are not up to 18 years do not have the legal autonomy to seek abortions even if they wish to do so. It is therefore, imperative that for such a vulnerable group not to be misinformed, they need to be empowered with knowledge on the safety of abortion from an early age [26]. In this sense, adolescence would be the ideal time to educate them on abortion related issues [27]. We acknowledge the fact that sexual health is already a vital component in the Ghanaian school curriculum, despite this, there may be cultural and religious barriers in discussing these topics at school.

In this current study, we noted regional disparities in self-efficacy in abortion decision-making among adolescent girls and young women in Ghana. Specifically, those in the Ashanti Region had the highest likelihood of taking decisions on their own to get an abortion whiles those from the Eastern Region were less likely to have self-efficacy in abortion decision-making. This is consistent with the results obtained by Owoo, Lambon-Quayefio, and Onuoha [9], where women in the Eastern and Ashanti Regions were less and more likely respectively to have self-efficacy in abortion decision-making. This resonates the explanation given by Darteh et al. [24] and GSS et al. [28] that regional differences exist in Ghana by level of education, development and poverty. The Ashanti Region is one of the regions that is well endowed with resources and the level of development compared to other regions is higher. Apart from these differences, there are also enormous cultural differences regarding sexual and reproductive health regionally. All these nuances might explain the disparities in decision-making efficacy on abortion among adolescent girls and young women in Ghana. It is therefore imperative that a qualitative study be carried out to explore the reasons surrounding abortion decision-making process and its self-efficacy. Future studies should also explore the impact of, particularly, the cultural specificity in relation to reproductive health decision making in general. Such studies would clarify our understanding on the phenomenon.

Relatedly, ethnicity was also associated with self-efficacy in abortion decision-making. The likelihood of self-efficacy in abortion decision-making was low among those belonging to the Ga-Adangbe ethnic group and this was different from what was found by Owoo, Lambon-Quayefio, and Onuoha [9]. The possible explanation for the differences in findings could be how ethnicity was grouped in the various studies as well as the differences in sample size and study populations.

## Strength and limitations

Whereas the findings from this study provide important considerations for advocacy and policy implications, some important limitations are worth acknowledging. First, the study shares all the weakness inherent in cross-sectional study design. Secondly, there is also the possibility of social desirability responses since the variables in the study were self-reported. Despite these weaknesses, the study has compelling strengths such as the relatively large sample size as well as the nationwide nature of the data which makes the results from the study generalizable to all adolescent girls and young women in Ghana.

## Conclusion

W used data from the 2017 GMHS to examine the association between mass media exposure and self-efficacy in abortion decision-making among adolescent girls and young women in

Ghana. Only a quarter of adolescent girls and young women in Ghana have self-efficacy in abortion decision-making. After controlling for other covariates, exposure to mass media was strongly associated with self-efficacy in abortion decision-making. In addition, age, marital status, ethnicity and region of residence were associated with self-efficacy in abortion decision-making. It is therefore recommended that to promote safe abortions, these factors should be taken into consideration to provide culturally and age appropriate support systems and education via the various media platforms to help adolescent girls and young women who might desire to have an abortion. Government should also ensure regular, periodic mass media campaigns to target adolescent girls and young women and provide education/knowledge on family planning and safe abortion practices. This could go a long way to ensure that cases of unsafe abortion are reduced to the starkest minimum and will go a long way to help in the reduction in maternal mortality cases.

## Acknowledgments

We acknowledge Measure DHS for providing us with the data upon which the findings of this study were based. We also acknowledge Mr. Ebenezer Agbaglo of the Department of English, University of Cape Coast, who thoroughly copy-edited this manuscript for language usage, spelling and grammar.

## Author Contributions

**Conceptualization:** Bright Opoku Ahinkorah, Abdul-Aziz Seidu, Georgina Yaa Mensah, Eugene Budu.

**Data curation:** Bright Opoku Ahinkorah, Abdul-Aziz Seidu, Georgina Yaa Mensah, Eugene Budu.

**Formal analysis:** Bright Opoku Ahinkorah, Abdul-Aziz Seidu, Georgina Yaa Mensah, Eugene Budu.

**Funding acquisition:** Bright Opoku Ahinkorah, Abdul-Aziz Seidu, Georgina Yaa Mensah, Eugene Budu.

**Investigation:** Bright Opoku Ahinkorah, Abdul-Aziz Seidu, Georgina Yaa Mensah, Eugene Budu.

**Methodology:** Bright Opoku Ahinkorah, Abdul-Aziz Seidu, Georgina Yaa Mensah, Eugene Budu.

**Project administration:** Bright Opoku Ahinkorah, Abdul-Aziz Seidu, Georgina Yaa Mensah, Eugene Budu.

**Resources:** Bright Opoku Ahinkorah, Abdul-Aziz Seidu, Georgina Yaa Mensah, Eugene Budu.

**Software:** Bright Opoku Ahinkorah, Abdul-Aziz Seidu, Georgina Yaa Mensah, Eugene Budu.

**Supervision:** Bright Opoku Ahinkorah, Abdul-Aziz Seidu, Georgina Yaa Mensah, Eugene Budu.

**Validation:** Bright Opoku Ahinkorah, Abdul-Aziz Seidu, Georgina Yaa Mensah, Eugene Budu.

**Visualization:** Bright Opoku Ahinkorah, Abdul-Aziz Seidu, Georgina Yaa Mensah, Eugene Budu.

**Writing – original draft:** Bright Opoku Ahinkorah, Abdul-Aziz Seidu, Georgina Yaa Mensah, Eugene Budu.

**Writing – review & editing:** Bright Opoku Ahinkorah, Abdul-Aziz Seidu, Georgina Yaa Mensah, Eugene Budu.

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
