## [Decision Letter · Decision Letter 0]

19 Aug 2020

PONE-D-20-18542

Mass media exposure and self-efficacy in abortion decision-making among young women in Ghana: Analysis of the 2017 Maternal Health Survey

PLOS ONE

Dear Dr. Budu,

Thank you for submitting your manuscript to PLOS ONE. After careful consideration, we feel that it has merit but does not fully meet PLOS ONE’s publication criteria as it currently stands. Therefore, we invite you to submit a revised version of the manuscript that addresses the points raised during the review process.

SPECIFIC ACADEMIC EDITOR COMMENTS: Two experts in the field handled your manuscript. We are very thankful for their expertise and efforts. Although interest was found in your study, there were some comments raised that overshadowed the current enthusiasm. These comments relate to the need for the authors to clarify and elaborate on several statements throughout the manuscript. There is also the need for additional analyses, including the dose response of the frequency of media exposure on self-efficacy in abortion decision making and a cross tabulation of the media exposure frequency variable on the outcomes. Please address ALL of the reviewers' comments in your revised manuscript.

We look forward to receiving your revised manuscript.

Kind regards,

Frank T. Spradley

Academic Editor

PLOS ONE

Journal Requirements:

https://reproductive-health-journal.biomedcentral.com/articles/10.1186/s12978-019-0775-9

https://www.ncbi.nlm.nih.gov/pmc/articles/PMC4248563/

https://link.springer.com/article/10.1186/s12905-019-0759-5?code=8415f13f-cf5d-4b52-bc4d-d06a34b2cf9a

https://www.mdpi.com/1660-4601/15/2/329/html

In your revision ensure you cite all your sources (including your own works), and quote or rephrase any duplicated text outside the methods section. Further consideration is dependent on these concerns being addressed.

Reviewers' comments:

Reviewer's Responses to Questions

**Comments to the Author**

1. Is the manuscript technically sound, and do the data support the conclusions?

Reviewer #1: Yes

Reviewer #2: Yes

2. Has the statistical analysis been performed appropriately and rigorously? 

Reviewer #1: Yes

Reviewer #2: Yes

3. Have the authors made all data underlying the findings in their manuscript fully available?

Reviewer #1: Yes

Reviewer #2: Yes

4. Is the manuscript presented in an intelligible fashion and written in standard English?

Reviewer #1: Yes

Reviewer #2: Yes

5. Review Comments to the Author

Reviewer #1: Reviewer Comments for Authors:

Title: Mass media exposure and self-efficacy in abortion decision-making among young women in Ghana: Analysis of the 2017 Maternal Health Survey

Manuscript Number: PONE-D-20-18542

General comments: The topic is very interesting and the manuscript also well written except the following comments which needs correction.

Abstract Part:

1. The abstract part was structurally well written but there is some errors:

For example under introduction part of abstract line number 48, “we sought to explore the association between mass media exposure and young women’s self-efficacy in abortion decision making.” Since your research is based on the data from survey and it is quantitative data, you can’t say to explore….. because this term is best term when we conduct qualitative research on new research ideas. Better to change this term.

2. Under materials and methods parts of abstract you said “Binary Logistic Regression models were employed with statistical significance pegged at p-value 56 <0.05.” binary logistic regression was not final model it is better if you change to ‘multivariable logistic regression model’.

3. The conclusion part under abstract is too large, better if summarized shortly and there was recommendation under conclusion, you should add recommendation term on sub topic conclusion or delete recommendations.

Introduction Part

1. The introduction part is good and well written but too large so better to summarize to maximum one and half page.

2. On page 4 lines no 100 “According to [4], 2.2 million unplanned pregnancies…. Not clear?

Materials and Methods Part

The Materials and Methods Part was well written except some limitations:

1. On page 8 line number 192, you said “Any respondent who selected at least one ‘yes’ for all the four was considered as exposed to mass media” Do you think exposure to one mass media can create self-efficacy on abortion decision making?

2. How you defined/measured self-efficacy in abortion decision-making in your study?

3. On page 8 line 216-217 you said “Binary Logistic Regression models were employed with statistical significance pegged at p-value 56 <0.05.” binary logistic regression was not final model it is better if you change to ‘multivariable logistic regression model’ or justify your reason.

Result Part

1. On page 9 lines 235-237, you said, “Figure 1 presents results of the proportion of self-efficacy in abortion decision-making among young women in Ghana. As shown in Figure 1, less than a quarter of young women (24%) in Ghana could decide on their own to get an abortion.” Unnecessary description….. Correct like this… “Less than a quarter of young women (24%) in Ghana could decide on their own to get an abortion.(Figure 1)”

2. On page 9 line 246-247 “Table 1 presents results of the self-efficacy in abortion decision making of young women 247 across mass media exposure and socio-demographic characteristics.” .. Delete it because it is Unnecessary description and add (Table 1) at the end of table I description. Do the same correction for all figures and tables in the result sections.

Discussion Part

1. Page 14 lines 309-311 -----“This current study also showed that young women who are exposed to mass media are more likely to make decisions on their own should they wish to have an induced abortion. Grammatically not correct.

2. Page 14 line 315-317------“The probable explanation they gave to this association was that women who are exposed to mass media are more likely to be exposed to “western ideas and messages that encourage independence and greater autonomy”. For whom this idea belongs? Study participants or…?

Conclusion part

It is good and well written; the study objectives were not concluded.

References Part

The reference numbers 5,10,11,16,18,23,24,30 and 34 were intolerably Too old references. So please replace with up to date references.

Reviewer #2: Abstract

Introduction:

1. Please provide some few quantifications of the epidemiology of abortion rates globally and in LMIC (magnitude of the problem).

Materials and methods

2. It may be better to include a statement concerning utilisation of crude and adjusted logistic regression models.

3. How was self-efficacy in abortion decision making measured?

Conclusion

4. Recommendations need to be action oriented. May be better to mention specifically who should be implementing them, and how. E.g. We recommend promotion of mass media campaigns scheduled on regular intervals in order to inform the target audience about safe abortions in Ghana.

Introduction

1. Great job at the introduction, it is well explained.

2. Would have also been good to elaborate on the cultural aspects and context around abortion in Sub Saharan Africa/ Ghana and how it affects autonomy in decision making for women who seek it. Is seeking an abortion something young women would openly talk about?

3. Would also be good to give examples of media campaigns in Ghana which had targeted messaging on abortion (With their citations).

4. Is the media exposure measured in the study as a general exposure? Or rather exposure on “Abortion services and laws” in mass media exposure?

Methods

1. The explanation of the dataset and sample section was well put-together, I suggest including a flow diagram as well to make it easier for the readers, but leave it up to the authors to decide on this.

2. Was there any basis/previous study that guided the coding of media exposure? How was the decision arrived to categorise a woman as exposed to media if she answered yes to at least one of the four media questions? (any previous literature?)

Results

1. Would have also been interesting to see the dose response of the frequency of media exposure on self-efficacy in abortion decision making, (i.e are women who are more frequently exposed to media have higher/lower self-efficacy?). A cross tabulation of the media exposure frequency variable on the outcome would be informative to show, while keeping the exposure as binary in logistic regression analysis.

Conclusion

1. How about suggesting, regular, periodic mass media campaigns to target young women and provide education/knowledge on family planning and safe abortion practices?

6. PLOS authors have the option to publish the peer review history of their article (what does this mean?). If published, this will include your full peer review and any attached files.

Reviewer #1: No

Reviewer #2: No

---

## [Author Response · Author response to Decision Letter 0]

26 Aug 2020

AUTHOR’S RESPONSE TO REVIEWS

Title: Mass media exposure and self-efficacy in abortion decision-making among adolescent girls and young women in Ghana: Analysis of the 2017 Maternal Health Survey

***Please link our ORCID IDs***

Authors names, email addresses and ORCID IDs

BOA: brightahinkorah@gmail.com(0000-0001-7415-895X)

AS: abdul-aziz.seidu@stu.ucc.edu.gh (0000-0001-9734-9054)

GYS: yaamensahgina@gmail.com

EB: budueugene@gmail.com(0000-0002-8484-9225)

Version:1

Date: 26/8/2020

Manuscript ID: PONE-D-20-18542

The Editor

PLOS ONE

26/8/2020

Dear Editor and Reviewers,

On behalf of all authors, I convey our gratitude to you for the critical and constructive review that has led to the improvement of our paper entitled “Mass media exposure and self-efficacy in abortion decision-making among adolescent girls and young women in Ghana: Analysis of the 2017 Maternal Health Survey”. We have revised the manuscript based on the comments raised by both reviewers. We believe the manuscript has improved substantively and will be published in your reputable journal, PLOS ONE. All the changes have been marked yellow in the revised manuscript. Please address all correspondence to me via email at: budueugene@gmail.com

Thank you. 

Yours Sincerely,

Eugene Budu

(Corresponding author)

REVIEWER REPORTS

REVIEWER 1:

Title: Mass media exposure and self-efficacy in abortion decision-making among young women in Ghana: Analysis of the 2017 Maternal Health Survey

Manuscript Number: PONE-D-20-18542

General comments 

The topic is very interesting and the manuscript also well written except the following comments which needs correction.

Abstract Part:

1. Comment: The abstract part was structurally well written but there is some errors:

For example under introduction part of abstract line number 48, “we sought to explore the association between mass media exposure and young women’s self-efficacy in abortion decision making.” Since your research is based on the data from survey and it is quantitative data, you can’t say to explore….. because this term is best term when we conduct qualitative research on new research ideas. Better to change this term.

Response: We have changed explore to examine. See page 2 line 50. 

2. Comment: Under materials and methods parts of abstract you said “Binary Logistic Regression models were employed with statistical significance pegged at p-value 56 <0.05.” binary logistic regression was not final model it is better if you change to ‘multivariable logistic regression model’.

Response: We thank the reviewer for this comment and suggestion. Notwithstanding, Binary Logistic Regression comprises both bivariate and multivariable logistic regression models. As seen in Table 2, the bivariate model was presented in Model I and multivariable model in Model II. For both models, p-value was considered as significant at <0.05.

3. Comment: The conclusion part under abstract is too large, better if summarized shortly and there was recommendation under conclusion, you should add recommendation term on sub topic conclusion or delete recommendations.

Response: We thank the reviewer for this comment and suggestion. We have summarised the conclusion. However, in relation to the recommendations, we consider that as an important aspect of the conclusion. However, the format for the journal is such that recommendations are considered part of the content of the conclusion. Therefore, the journal format for sections under abstract will not allow us to add “recommendation” to the sub-topic conclusion. Page 3 line 75-81. 

Introduction Part

4. Comment: The introduction part is good and well written but too large so better to summarize to maximum one and half page.

Response: We thank the reviewer for this comment and suggestion. We have summarized the background to 1.5 pages. See page 4-5. Line 95-134. 

5. Comment: On page 4 lines no 100 “According to [4], 2.2 million unplanned pregnancies…. Not clear?

Response: We have made this sentence clear. 

Materials and Methods Part

6. Comment: The Materials and Methods Part was well written except some limitations:

On page 8 line number 192, you said “Any respondent who selected at least one ‘yes’ for all the four was considered as exposed to mass media” Do you think exposure to one mass media can create self-efficacy on abortion decision making?

Response: Yes. exposure to at least one media can create self-efficacy. This is because exposure just one media source can give so much information on abortion which can enhance self-efficacy. 

7. Comment: How you defined/measured self-efficacy in abortion decision-making in your study?

Response: We thank the reviewer for this comment. Self-efficacy in abortion decision-making derived from the question ‘Could you decide on your own to get an abortion?. Those who answered “Yes” to this question were considered as having self-efficacy in abortion decision making whiles those who responded “No” to the question were considered as not having self-efficacy in abortion decision making. See page 160-165. 

8. Comment: On page 8 line 216-217 you said “Binary Logistic Regression models were employed with statistical significance pegged at p-value 56 <0.05.” binary logistic regression was not final model it is better if you change to ‘multivariable logistic regression model’ or justify your reason.

Response: We thank the reviewer for this comment and suggestion. Notwithstanding, Binary Logistic Regression comprises both bivariate and multivariable logistic regression models. As seen in Table 2, the bivariate model was presented in Model I and multivariable model in Model II. For both models, p-value was considered as significant at <0.05.

Result Part

9. Comment: On page 9 lines 235-237, you said, “Figure 1 presents results of the proportion of self-efficacy in abortion decision-making among young women in Ghana. As shown in Figure 1, less than a quarter of young women (24%) in Ghana could decide on their own to get an abortion.” Unnecessary description….. Correct like this… “Less than a quarter of young women (24%) in Ghana could decide on their own to get an abortion.(Figure 1).”

Response: We have corrected this section per the suggestion of the reviewer.

10. Comment: On page 9 line 246-247 “Table 1 presents results of the self-efficacy in abortion decision making of young women 247 across mass media exposure and socio-demographic characteristics.” .. Delete it because it is Unnecessary description and add (Table 1) at the end of table I description. Do the same correction for all figures and tables in the result sections.

Response: We have corrected these sections per the suggestion of the reviewer.

Discussion Part

11. Comment: Page 14 lines 309-311 -----“This current study also showed that young women who are exposed to mass media are more likely to make decisions on their own should they wish to have an induced abortion. Grammatically not correct.

Response: We have corrected the sentence to read “This study also showed that young women who are exposed to mass media are more likely to have self-efficacy to make abortion decisions.” See page 13 line 286-288. 

12. Comment: Page 14 line 315-317------“The probable explanation they gave to this association was that women who are exposed to mass media are more likely to be exposed to “western ideas and messages that encourage independence and greater autonomy”. For whom this idea belongs? Study participants or…?

Response: We have clarified this section of the paper. Page 13 line 292-294. 

Conclusion part

13. Comment: It is good and well written; the study objectives were not concluded.

Response: We have added the study objective to the conclusion. See page 16 line 351-353. 

References Part

14. Comment: The reference numbers 5,10,11,16,18,23,24,30 and 34 were intolerably Too old references. So please replace with up to date references.

Response: We have taken out all the old references. Some have been replaced whiles others have been taken out in the course of reducing the length of the introduction. 

REVIEWER #2: 

Abstract

Introduction:

15. Comment: Please provide some few quantifications of the epidemiology of abortion rates globally and in LMIC (magnitude of the problem).

Response: We have added abortions rates globally, in LMIC, SSA and Ghana to the background. See page 2 line 46-47. 

Materials and methods

16. Comment: It may be better to include a statement concerning utilisation of crude and adjusted logistic regression models.

Response: We have added a statement concerning utilisation of crude and adjusted logistic regression models. Page 2 line 61-62. 

17. Comment: How was self-efficacy in abortion decision making measured?

Response: We have included how self-efficacy in abortion decision making was measured in the methods. See page 2 line 56-59. 

Conclusion

18. Comment: Recommendations need to be action oriented. May be better to mention specifically who should be implementing them, and how. E.g. We recommend promotion of mass media campaigns scheduled on regular intervals in order to inform the target audience about safe abortions in Ghana.

Response: We thank the reviewer for the suggestion. We have revised the recommendation appropriately. It now reads “We recommend that policy makers should promote mass media campaigns scheduled on regular intervals in order to inform the target audience about safe abortions in Ghana. This could go a long way to ensure that cases of unsafe abortions are reduced to the starkest minimum”. See page 3, line 78-81

Introduction

19. Comment: Great job at the introduction, it is well explained.

Would have also been good to elaborate on the cultural aspects and context around abortion in Sub Saharan Africa/Ghana and how it affects autonomy in decision making for women who seek it. Is seeking an abortion something young women would openly talk about? Would also be good to give examples of media campaigns in Ghana which had targeted messaging on abortion (With their citations).

Response: We thank the reviewer for the commendation and suggestions. We have revised the background to incorporate the suggestions. See page 4, line 101-105

20. Comment: Is the media exposure measured in the study as a general exposure? Or rather exposure on “Abortion services and laws” in mass media exposure?

Response: In this study, media exposure is measured as a general exposure. We assume that those who are exposed could be exposed to information on abortion through their exposure to media. 

Methods

21. Comment: The explanation of the dataset and sample section was well put-together, I suggest including a flow diagram as well to make it easier for the readers, but leave it up to the authors to decide on this.

Response: We thank the reviewer for the suggestion on the flow diagram. However, the authors are more comfortable writing the methods under sub-sections rather than using a flow diagram. 

22. Comment: Was there any basis/previous study that guided the coding of media exposure? How was the decision arrived to categorise a woman as exposed to media if she answered yes to at least one of the four media questions? (any previous literature?)

Response: We have provided reference to support our categorisation. See page 7 line 179.

Results

23. Comment: Would have also been interesting to see the dose response of the frequency of media exposure on self-efficacy in abortion decision making, (i.e are women who are more frequently exposed to media have higher/lower self-efficacy?). A cross tabulation of the media exposure frequency variable on the outcome would be informative to show, while keeping the exposure as binary in logistic regression analysis.

Response: We thank the reviewer for this suggestion. Unfortunately, we are unable to create categories on the basis of frequency from the four media sources to measure assess the frequency of media exposure on self-efficacy in abortion decision making. This is because, whereas exposure to radio, newspaper and television had three responses (almost every day, at least once a week, less than once a week or not at all), exposure to internet was coded as “yes” and “No” in the dataset. This makes it impossible to create a string variable from four variables unless dichotomous variable as we created.

Conclusion

24. Comment: How about suggesting, regular, periodic mass media campaigns to target young women and provide education/knowledge on family planning and safe abortion practices?

Response: We thank the reviewer for the suggestion. We have added this to the conclusion. See page 16 line 360-362

---

## [Decision Letter · Decision Letter 1]

16 Sep 2020

Mass media exposure and self-efficacy in abortion decision-making among adolescent girls and young women in Ghana: Analysis of the 2017 Maternal Health Survey

PONE-D-20-18542R1

Dear Dr. Budu,

We’re pleased to inform you that your manuscript has been judged scientifically suitable for publication and will be formally accepted for publication once it meets all outstanding technical requirements.

Kind regards,

Frank T. Spradley

Academic Editor

PLOS ONE

Reviewers' comments:

Reviewer's Responses to Questions

**Comments to the Author**

1. If the authors have adequately addressed your comments raised in a previous round of review and you feel that this manuscript is now acceptable for publication, you may indicate that here to bypass the “Comments to the Author” section, enter your conflict of interest statement in the “Confidential to Editor” section, and submit your "Accept" recommendation.

Reviewer #1: All comments have been addressed

Reviewer #2: All comments have been addressed

2. Is the manuscript technically sound, and do the data support the conclusions?

Reviewer #1: Yes

Reviewer #2: Yes

3. Has the statistical analysis been performed appropriately and rigorously? 

Reviewer #1: Yes

Reviewer #2: Yes

4. Have the authors made all data underlying the findings in their manuscript fully available?

Reviewer #1: Yes

Reviewer #2: Yes

5. Is the manuscript presented in an intelligible fashion and written in standard English?

Reviewer #1: Yes

Reviewer #2: Yes

6. Review Comments to the Author

Reviewer #1: Reviewer Comments for Authors:

Title: Mass media exposure and self-efficacy in abortion decision-making among young women in Ghana: Analysis of the 2017 Maternal Health Survey

Manuscript Number: PONE-D-20-18542R1

General Comments: I am with response given for my suggestion and questions on the given manuscript. The manuscript was well revised based on the given comments. All my questions and unclear points were addressed in the revised manuscript.

Specific Comments: I don’t think exposure to a single media can create self-efficacy on abortion decision making. Even the exposure was general exposure not specific abortion related exposure. I am not convinced by your response on this issue.

Reviewer #2: Well done authors and a good job! All the comments have been very well addressed.

7. PLOS authors have the option to publish the peer review history of their article (what does this mean?). If published, this will include your full peer review and any attached files.

Reviewer #1: **Yes: **Biru Abdissa Mizana

Reviewer #2: No

---

## [Editor Report · Acceptance letter]

18 Sep 2020

PONE-D-20-18542R1 

Mass media exposure and self-efficacy in abortion decision-making among adolescent girls and young women in Ghana: Analysis of the 2017 Maternal Health Survey 

Dear Dr. Budu:

I'm pleased to inform you that your manuscript has been deemed suitable for publication in PLOS ONE. Congratulations! Your manuscript is now with our production department. 

Kind regards, 

on behalf of

Dr. Frank T. Spradley 

Academic Editor

PLOS ONE